# Mental Health and Health-Related Quality of Life of Children and Youth during the First Year of the COVID-19 Pandemic: Results from a Cross-Sectional Survey in Saskatchewan, Canada

**DOI:** 10.3390/children10061009

**Published:** 2023-06-03

**Authors:** Nazeem Muhajarine, Vaidehi Pisolkar, Tamara Hinz, Daniel A. Adeyinka, Jessica McCutcheon, Mariam Alaverdashvili, Senthil Damodharan, Isabelle Dena, Christa Jurgens, Victoria Taras, Kathryn Green, Natalie Kallio, Yolanda Palmer-Clarke

**Affiliations:** 1Saskatchewan Population Health and Evaluation Research Unit (SPHERU), University of Saskatchewan, 104 Clinic Place, Saskatoon, SK S7N 2Z4, Canada; vaidehi.pisolkar@usask.ca (V.P.); daniel.adeyinka@usask.ca (D.A.A.); isabelle.dena@usask.ca (I.D.); yolanda.palmer@usask.ca (Y.P.-C.); 2Department of Community Health and Epidemiology, College of Medicine, University of Saskatchewan, 107 Wiggins Rd, Saskatoon, SK S7N 5E5, Canada; 3Department of Psychiatry, College of Medicine, University of Saskatchewan, 103 Hospital Drive, Saskatoon, SK S7N 0W8, Canada; tamara.hinz@usask.ca (T.H.); mariam.alaverdashvili@usask.ca (M.A.); skd357@mail.usask.ca (S.D.); 4Saskatchewan Health Authority, 701 Queen Street, Saskatoon, SK S7K 0M7, Canada; 5Canadian Hub for Applied and Social Research (CHASR), University of Saskatchewan, 9 Campus Drive, Saskatoon, SK S7N 5A5, Canada; jessica.mccutcheon@usask.ca; 6EGADZ Saskatoon Downtown Youth Centre Inc., 1st Avenue North, Saskatoon, SK S7K 1X5, Canada; 7Saskatchewan Teachers’ Federation, 2317 Arlington Avenue, Saskatoon, SK S7J 2H8, Canada

**Keywords:** COVID-19, anxiety, depression, health-related quality of life, children and youth, equity

## Abstract

For children and youth, the COVID-19 pandemic surfaced at a critical time in their development. Children have experienced extended disruptions to routines including in-person schooling, physical activities, and social interactions—things that bring meaning and structure to their daily lives. We estimated the prevalence of anxiety and depression symptoms of children and youth and their experiences of health-related quality of life (HRQoL), during the first year of the pandemic, and identified factors related to these outcomes. Further, we examined these effects among ethnocultural minority families. We conducted an online survey (March–July 2021) with 510 children and youth aged 8–18 years and their parents/caregivers. The sample was representative of the targeted population. We modelled the relationship between anxiety, depression (measured using the Revised Child Anxiety and Depression Scale), HRQoL (measured using KIDSCREEN-10), and sociodemographic, behavioural, and COVID-19-contributing factors using binary logistic regression. A priori-selected moderating effects of sociodemographic characteristics and self-identified ethnocultural minority groups on the outcomes were tested. The point-in-time prevalence of medium-to-high anxiety symptoms and depression symptoms was 10.19% and 9.26%, respectively. Almost half (49.15%) reported low-to-moderate HRQoL. Children reporting medium-to-high anxiety symptoms, depression symptoms, and low-to-moderate HRQoL were more likely to be aged 8–11 years, 16–18 years, ethnocultural minority participants, living in rural/urban areas, having good/fair MH before COVID-19, experiencing household conflicts, having less physical activity, and having ≥3 h of recreational screen time. Those who had more people living at home and ≥8 h of sleep reported low anxiety and depression symptoms. Ethnocultural minority 16–18-year-olds were more likely to report low-to-moderate HRQoL, compared to 12–15-year-olds. Additionally, 8–11-year-olds, 16–18-year-olds with immigrant parents, and 16–18-year-olds with Canadian-born parents were more likely to report low–moderate HRQoL, compared to 12–15-year-olds. Children and youth MH and HRQoL were impacted during the pandemic. Adverse MH outcomes were evident among ethnocultural minority families. Our results reveal the need to prioritize children’s MH and to build equity-driven, targeted interventions.

## 1. Introduction

Mounting evidence suggests humanitarian emergencies and disease outbreaks are a threat to social cohesion. Risks for poor mental health (MH) are heightened during such times and may persist over time [1,2]. The broader impacts of the COVID-19 pandemic on children and youth merit special attention to mitigate potential long-lasting consequences on their MH and wellbeing. For children and youth, the COVID-19 pandemic surfaced at a critical time in their development. Children have experienced extended disruptions to routines including in-person schooling, physical activities, and social interactions—things that bring meaning and structure to their daily lives [3,4,5,6]. Furthermore, in many countries, access to MH interventions and services was entirely or partially disrupted, including school MH programs that children heavily rely on for support [7].

A meta-analysis reported a dramatic increase in the global prevalence of anxiety (25.2%) and depression (20.5%) in children and adolescents since the pandemic [8]. A study conducted in Britain on 168 children with a mean age of 10 years revealed a moderate to substantial rise in depression based on the Revised Child Anxiety and Depression Scale (RCADS) score [9]. Just before the pandemic, a 2019 Canadian Health Survey on Children and Youth reported that about 4% of children and youth (1–17 years of age) had fair or poor MH [10]. A Canadian study during the pandemic identified that 67–70% of children (6–18 years of age) have reported a decline in a minimum of one MH outcome while 19–31% of children showed improvement in one MH outcome [11]. A decreased health-related quality of life (HRQoL) was also evident compared to before the pandemic [12].

The framework guiding our analysis is the Commission on Social Determinants of Health (CSDH) framework that explains how the different social determinants of health influence health outcomes and health inequities across populations [13]. Equity in health refers to the “absence of systematic disparities in health (or in the major social determinants of health) between groups with different levels of underlying social advantage/disadvantage—that is, wealth, power, or prestige” [14].

During the pandemic, research showed that adverse MH outcomes were more severe among girls, older children, and those with pre-existing MH conditions [8,11,15]. Studies found that those who experienced an economic crisis during COVID-19 reported an increased link with mental health symptoms [16,17]. Early studies during the pandemic reported an adverse impact on children’s behaviours including sleep patterns, screen time, and physical activity [11,18,19,20,21,22]. According to a study carried out in 2021 in Austria on 2290 participants aged 6–18 years, there was a positive correlation observed between pandemic-related anxiety, sleeping problems, and poor sleep quality [20]. With most schools moving to remote learning, screen time increased, and higher screen use contributed to the negative impact on MH [11,22]. On the other hand, early research conducted during the pandemic indicated that participating in physical activity had a protective effect on mental health outcomes [21,23,24]. Adopting remote learning was another factor contributing to adverse MH outcomes. In 2020, a study conducted on 567 adolescents in grades 7–12 revealed that those who received virtual schooling exhibited poorer mental health compared to students who received in-person schooling [25].

Although studies have identified the prevalence of MH difficulties in children during the pandemic, there is still a need for child-reported data and an understanding of the mechanisms and processes that influence MH [26]. Additionally, the impact of COVID-19 and governmental response to it differed across Canadian provinces and territories. Therefore, a comprehensive understanding of the MH impact on children and youth in each jurisdiction is warranted.

Our primary objective was to identify factors that increase or decrease risk across three MH domains: anxiety, depression, and health-related quality of life in children (defined as 8–15-year-olds) and youth (16–18-year-olds) in Saskatchewan—to coincide with the end of the first full academic year during COVID-19. The second objective was to examine whether these impacts were worsened or buffered among equity-deserving groups (those who experience structural and systemic marginalization, such as minority and racialized communities). We analysed how age, gender, and income intersect with ethnocultural status (ethnicity and immigration status).

## 2. Materials and Methods

### 2.1. Brief Timeline of COVID-19 Measures, and Schools in Saskatchewan

Appendix A show the temporal pattern of COVID-19 incidence and a timeline of pandemic response in elementary and secondary schools in Saskatchewan. The first presumptive COVID-19 case was announced on 12 March 2020 [27]. On March 20, schools closed for in-person learning. The province announced remote learning for schools on March 30. Further, the Saskatchewan government released its Safe Schools Plan for 2020–2021 on August 4, with the reopening of schools on 1 September 2020. When this study’s survey opened to children, youth, and families in March 2021, there was a downward trend in COVID-19 incidence rates across the province. COVID-19 cases rose slowly in April and started declining in May and throughout the summer. This was also when vaccination for teachers began (May 4), and round 2 of Saskatchewan’s “reopening” commenced. By the end of May 2021, the Saskatchewan Health Authority began vaccination for children aged 12 years and older in schools [28]. All public health orders were lifted in July 2021.

### 2.2. Study Sample and Design

This cross-sectional study consisted of 510 children aged 8–18 years and their parents/caregivers. Data were collected between 19 March and 27 July 2021. The data collection phase coincided with the phased re-opening across Saskatchewan. Using 2016 Census data, a ‘sample frame’ of Saskatchewan households with children 5–19 years old, stratified by age, gender, and location, was created to reflect this population distribution. Participants were recruited from all school divisions in Saskatchewan, including public, Catholic, independent, and First Nation-administered schools. Participants were also included from an online representative Community Panel (recruited initially through random selection), managed by the Canadian Hub of Applied and Social Research, a consortium of research labs at the University of Saskatchewan. Participants completed online surveys. The child/youth first completed the self-report questionnaires, followed by their parent/caregiver. Parents completed some key demographic questions. To ensure representativeness, we weighted the samples by age, gender, and location in Saskatchewan using the 2016 Canadian Census data.

### 2.3. Survey Development

The survey was developed through a collaborative approach with engagement from interdisciplinary researchers, psychiatrists, social workers, and a Parent and Community Advisory Council. Collaborators came from diverse experiences, expertise, and backgrounds and informed and guided the development of survey items. The Parent and Community Advisory Council consisted of parent–child pairs, including youth with lived experiences with MH challenges and a youth representative from our community partner. They provided significant insights into the development of the survey items.

### 2.4. Measures

#### 2.4.1. Dependent Variables

The outcome variables were (1) anxiety symptoms, (2) depression symptoms, and (3) health-related quality of life. Children and youth were asked to report their feelings and experience of symptoms in the past 2 weeks. MH conditions were measured using standardized and validated tools previously used in studies evaluating COVID-19 impact on MH [9,12,29]. Anxiety and depression symptoms were measured using the Revised Child Anxiety and Depression Scale (RCADS-25) and quality of life was measured using KIDSCREEN-10.

The 25-item RCADS (15 items measuring symptoms of anxiety and 10 items measuring symptoms of depression) is scored on a 4-point Likert scale from 0 = “Never” to 3 = “Always” [30,31,32]. The 10 items in KIDSCREEN-10 tap into physical, psychological, and social aspects of health-related quality of life (HRQoL) [33,34]. The items were presented with 5-point response scales, from 0 = “never” to 4 = “always” [12,33].

To derive prorated scores, responses from the respective items were summed. Further, the total scores were divided by the number of questions answered. The severity of anxiety and depression symptoms was categorized as low (0–64) [reference category] and moderate-to-high (>65) [32]. HRQoL was categorized as low-to-moderate (0–70) and high (71–100) [reference category]. The reliability (internal consistency) of anxiety, depression, and HRQoL was good (α = 0.87, 0.89, and 0.81, respectively) and it aligns with previous studies [33,35,36].

#### 2.4.2. Independent Variables

Sociodemographic, behavioural, and other COVID-19-contributing factors were studied (see Table 1). Physical activity was grouped in accordance with current 24 h movement guidelines [37]. Based on the literature, the categories for household density were created [38]. When the missing responses exceeded a threshold of 15%, these were labelled as missing and included in the models. For example, family income (24.19%), or if the child had not known COVID-19 cases in the classroom (34.43%), was labelled as missing and used in our analyses. Due to theoretical relevance of the association of sleep with HRQoL, sleep was retained in the final regression model [39,40].

### 2.5. Statistical Analyses

We modelled the relationship between anxiety symptoms, depression symptoms, and HRQoL, and relevant sociodemographic, behavioural, and other COVID-19-contributing factors. We tested a priori-selected interactions (age, gender, income with ethnicity, and parent immigration status). We conducted binary logistic regression. Bivariate analyses were tested using the Chi-square test or Fischer’s exact test (when expected frequency < 5) (Appendix A). Candidate variables for the regression were selected based on a *p*-value of <0.25. Multicollinearity for candidate variables was assessed by computing the mean variance inflation factor (VIF). We employed R-squared, the Bayesian Information Criterion (BIC), and the Akaike’s Information Criterion (AIC) to assess model performance. Adjusted odds ratios (aOR) and 95% confidence intervals (CI) were used to estimate the strength of the association. The level of statistical significance was set at *p* < 0.05. To maintain representativeness of data in the analysis stage, sample weights were applied. Bivariate analyses were conducted in IBM SPSS Version 28.0 [41]. The multivariable regression models were fitted in STATA™ version 17.0 [42].

## 3. Results

### 3.1. Sample Description

The demographic characteristics of the child–parent dyads in our study are shown in Table 2. The sample included 510 children and youth in three age groups: 8–11 years (46.71%), 12–15 years (34.12%), and 16–18 years (19.18%). Children in elementary grades (1–8) accounted for 65.29%, and high school (9–12) for 31.93%. Those who identified as a girl/woman were 43.72% and those who identified as a boy/man were 50.98%. Participants identifying as an ethnocultural minority (Black, Indigenous, or Person of colour) were 23.74%. Participants who self-identified as white were 65.61%; same as 66.0% in the Saskatchewan population who self-identify as having European ethnicity, according to the 2021 Canadian Census data [43]. In total, 14% of the families reported being immigrants. Most families had an income ≥$100,000 (39.76%) and resided in a city, i.e., Saskatoon/Regina (39.01%).

### 3.2. Prevalence of Anxiety, Depression, and Health-Related Quality of Life

At the end of the first year of the pandemic, the point-in-time prevalence of medium-to-high anxiety and depression symptoms was 10.19% and 9.26%, respectively—as reported by children and youth. Almost half (49.15%) reported having low-to-moderate HRQoL (Figure 1).

#### 3.2.1. Anxiety Symptoms

Age, income, ethnicity, location, MH before COVID-19, household density, and screen time were significantly associated with medium-to-high anxiety symptoms (Figure 2). The full regression model is presented in the Appendix A.

Respondents aged 8–11 years were 4.54 times more likely to report anxiety symptoms (95% CI: 1.33 to 15.47) compared to respondents aged 12–15 years. Those self-identifying as an ethnocultural minority were 43.25 times more likely to report anxiety symptoms (95% CI: 4.18 to 447.81) compared to their white counterparts. Respondents living in rural and urban areas were more likely to report anxiety symptoms [OR: 8.44 (95% CI: 2.04 to 34.89); and OR: 4.07 (95% CI: 1.07 to 15.46)] compared to those residing in mid-size towns. Children and youth who had fair/poor or very good/good MH before COVID-19 were more likely to report anxiety symptoms [OR: 18.49 (95% CI: 2.69 to 127.03); and OR: 6.18 (95% CI: 1.11 to 34.36)] compared to those reporting excellent MH before COVID-19. Children with 3 or more hours of recreational screen time were 3.07 times more likely to report anxiety symptoms (95% CI: 1.16 to 8.12) compared to those with less than 3 h of screen time. Children living in households with greater numbers of people (>1 person per bedroom) were 0.13 times less likely to report anxiety symptoms (95% CI: 0.04 to 0.42) compared to those living with fewer people (≤1 person/bedroom).

We found that white children aged 8–11 years were 4.53 times and those aged 16–18 years were 1.17 times more likely to report medium–high anxiety symptoms, compared to 12–15-year-old white participants (Figure 3A). White girls were 2.07 times more likely to report anxiety symptoms compared to white boys (Figure 3B).

#### 3.2.2. Depression Symptoms

Age, location, household density, household conflicts, screen time, and sleep were significantly associated with depression symptoms (Figure 4). The full regression model is presented in the Appendix A. Additionally, 8–11-year-olds were 3.48 times more likely to report depression symptoms (95% CI: 1.03 to 11.76), compared to 12–15-year-olds. Respondents living in rural and urban areas were more likely to report depression symptoms [OR: 6.60 (95% CI: 1.01 to 43.17); and OR: 6.18 (95% CI: 1.45 to 26.29)] than those living in mid-size towns. Children who experienced more household conflicts [OR: 10.84 (95% CI: 2.28–51.51)], compared to fewer, were more likely to report depression symptoms. Children who reported 3 or more hours of recreational screen time were 5.20 times more likely to experience depression symptoms (95% CI: 1.52 to 17.80), compared to those who had less than 3 h of screen time.

In addition, 16–18-year-olds were 0.19 times less likely to report depression symptoms (95% CI: 0.04 to 0.89) as compared to 12–15-year-olds. Respondents living in households with a greater number of people (>1 person per bedroom) were 0.06 times less likely to report depression symptoms (95% CI: 0.01 to 0.24), compared to those living with fewer people (≤1 person per bedroom). Those who obtained 8 or more hours of sleep were also less likely to report depression symptoms as compared to those who slept less than 8 h [OR: 0.07 (95% CI: 0.02 to 0.26)].

#### 3.2.3. Health-Related Quality of Life

Age, MH status before COVID-19, household conflicts, physical activity, and screen time were significantly associated with low-to-moderate HRQoL (Figure 5). The full regression model is presented in the Appendix A.

Youth aged 16–18 years were 5.88 times more likely to report low–moderate QoL (95% CI: 1.66 to 20.86), compared to 12–15-year-olds. Those reporting good/very good MH also reported low–moderate QoL [OR: 3.25 (95% CI: 1.56 to 6.79)] as compared to respondents reporting excellent MH before COVID-19. Children who experienced more household conflicts during COVID-19 [OR: 3.38 (95% CI: 1.46 to 7.85)], compared to fewer, were more likely to report low QoL. Children who saw a change in PA, i.e., who reported to be less physically active, were 3.83 times more likely to report low–moderate QoL (95% CI: 1.46 to 10.04) as compared to those who did not have any change in activity. Children who reported 3 or more hours of recreational screen time were 2.7 times more likely to experience low QoL (95% CI: 1.11 to 6.58) as compared to those who had less than 3 h of screen time.

Among ethnocultural minority participants, youth (16–18 years) were 5.89 times more likely to report low–moderate QoL, compared to those aged 12–15 years. Among white participants, those aged 16–18 years were 5.48 times and those aged 8–11 years were 1.24 times more likely to report low–moderate QoL, compared to 12–15-year-old white participants (Figure 6A). Among ethnocultural minority respondents, those from households with annual incomes <$100,000 were 6.82 times more likely to report low–moderate QoL, compared to ≥$100,000 (Figure 6B). Among children whose parents were both Canadian-born, 16–18-year-olds were 5.96 times more likely to report low–moderate QoL, compared to those aged 12–15 years. On the other hand, among children whose parents were immigrants (i.e., none or one of the parents born in Canada), 8–11-year-olds were 5.53 times and 16–18-year-olds were 3.60 times more likely to report low–moderate QoL, compared to those aged 12–15 years (Figure 6C).

## 4. Discussion

Our study estimates the mental health and health-related QoL of children and youth at the end of the first year of the COVID-19 pandemic in Saskatchewan, Canada. Further, our analysis reveals the factors contributing to inequitable distribution of MH outcomes in children and youth.

Children and youth experienced poor MH symptoms and HRQoL during the first year of the pandemic. Overall, anxiety, depression, and HRQoL outcomes were influenced by several sociodemographic, behavioural, and other contributing factors. Risk was particularly higher for younger children (8–11 years) and youth (16–18 years), children who self-identified as an ethnocultural minority, those who lived in rural and urban areas (vs. small towns), participants who had good/fair MH before COVID-19 (vs. excellent MH), participants who experienced more household conflicts, those who had less PA, and those who had ≥3 h of recreational screen time. Girls who identified as white were more likely to report medium–high anxiety symptoms. White children aged 8–11 years, and ethnocultural minority and white youth aged 16–18 years, were more likely to report low–moderate QoL. Ethnocultural minority children and those from relatively lower-income families (<$100,000) were more likely to report low–moderate QoL. Among respondents whose parents were Canadian-born, 16–18-year-olds were more likely to report low-to-moderate QoL. For respondents whose parents were immigrants, 8–11-year-olds and 16–18-year-olds were more likely to report low-to-moderate QoL. Surprisingly, we found that the likelihood of MH symptoms was lower for those who lived in higher density households (>1 person/bedroom). The likelihood was also lower for those who had ≥8 h of sleep.

Our MH symptom estimates compared to pre-pandemic, 2019, were higher—doubled–but were lower compared to reported studies in other parts of the world [8,10]. Our findings need to be interpreted carefully taking the COVID-19 context and methodology into consideration. The prevalence rates of MH symptoms were contextual to time and place. Since our survey was conducted towards the end of the school year—the majority enrolled in May and June 2021—children were likely looking forward to summer holidays, and public health restrictions were eased in the province during this time.

Our study found that 8–11-year-olds were more likely to report symptoms of anxiety and depression. Although 16–18-year-old youth were less likely to have depression symptoms, they were more likely to report low-to-moderate HRQoL. Children are developmentally different from adults. They may not have developed the skills to make decisions accurately, communicate, cope, and seek help when needed—putting them at greater risk. It is essential to strengthen MH response services to address age-specific needs [44].

We found respondents with fair/poor and good/very good MH before COVID-19, compared to excellent MH, were more likely to report anxiety and low–moderate QoL. However, the effect size of fair/poor MH before the pandemic was much larger (OR: 18.49, 95% CI: 2.69–127.03) than the effect size of children with very good/good MH before the pandemic (OR: 6.18, 95% CI: 1.11–34.36). This suggests there was a stepped, or patterned, relationship between MH before the pandemic and at the end of the first year of the pandemic. Given that the pandemic caused disruptions to essential MH services—with longer wait times and lack of in-person support [7]—not being able to receive help in a timely manner may have contributed to exacerbation of MH issues in children reporting pre-COVID-19 MH issues. Additionally, switching to online schooling reduced access to school counselling services—a primary source of help for children. The lack of available MH and counselling supports for rural citizens in Saskatchewan [45], and the rising number of COVID-19 cases in cities like Saskatoon and Regina [46] during that period, can help elucidate the observed higher MH symptoms in children and youth residing in both rural and urban areas compared to smaller towns. An important lesson from the pandemic is to be prepared to deliver continued MH support during public health crises.

It was expected that children and youth reporting increased household conflicts were also at increased risk for depression and low-to-moderate QoL. Family dynamics were significantly altered during the pandemic. Changes to parenting processes, relationships between members, MH of the parent/caregiver, and stress and uncertainty related to COVID-19 caused conflicts in households [47,48]. However, this is not a simple association; we found that children living in households with more ‘density’ (more people per bedroom) had a reduced likelihood of anxiety and depression. Early observations during the pandemic showed that social isolation increased symptoms of anxiety and depression [49]. Having more people in the household that the child can relate to may have eased feelings of isolation or loneliness; this may have provided increased opportunities for interactions, family activities, emotional support, and resiliency [50,51,52]. Studies are needed to fully understand family dynamics in the subsequent years of the pandemic.

Behavioural factors including screen time, physical activity, and sleep were affected by COVID-19. Moreover, maladaptive changes had a significant impact on MH symptoms. Increased duration of recreational screen time was associated with poorer MH symptoms. This is consistent with the previous literature [22]. In this study, recreational screen use was associated the strongest with depression. Screen time increased for many as a tool to cope with boredom and distress over physical activity. Studies have shown that screen use (videogames and video-calling) during the pandemic greatly increased due to the unparalleled virtual social participation it rendered [53,54].

Research also indicates that a decrease in physical activity is associated with poor QoL [55]—a finding consistent with our study. A decline in PA during COVID-19 may be attributed to limited opportunities to engage in outdoor and indoor activities [54]—considering Saskatchewan has long winters and due to prolonged COVID-19 mandates. At the same time, parents juggling working from home in addition to lack of childcare, household chores, and remote school learning may have led to relaxing screen time limits and PA routines. Nevertheless, previous research has shown that parental co-participation and supportive engagement are crucial in facilitating movement behaviours for children [54,56]. Focusing on promoting parental co-participation, accelerating community and school programs to re-connect children in activities, and re-establishing routines is imperative.

Adequate sleep is considered crucial for MH; however, throughout the pandemic, increased flexibility in daily routines, amplified dependence on media and devices, and heightened fear, anxiety, and uncertainty with the global health crisis have all played a role in fostering inconsistent sleep patterns and sleep troubles among the paediatric population [19,57,58]. Previous research has suggested longer sleep duration to be associated with emotional regulation (e.g., symptoms of anxiety and depression) [59]. Consistent with that, this study found obtaining 8 or more hours of sleep reduced the risk for depression. Further investigation through longitudinal research to examine these associations in greater depth is necessary. Taken together, these findings strongly call for interventions to promote healthy movement behaviours in children.

Ethnocultural minority children aged 8–11 years and youth aged 16–18 years were less likely to report anxiety. This was a surprising finding. According to a recent study conducted in Saskatchewan, adults under 30 years of age who identified as visible minorities were less likely to report poor MH outcomes [60]. Minority and racialized groups have demonstrated resourcefulness and resilience stemming from enduring a historical trajectory of conquering hardships and preserving community bonds [61]. During COVID-19, Indigenous communities in Canada came together and collectively found innovative ways to support one another by crafting their own face masks, sharing food hampers, and organizing activities [62,63]. Drawing on these lessons of resilience will help reinforce coping skills in vulnerable groups during uncertain times. Additionally, since white children and youth, especially girls, reported a higher likelihood of anxiety, MH interventions should be targeted.

Both white and ethnocultural minority 16–18-year-olds were more likely to report low-to-moderate quality of life, a finding consistent with research studies in Canada that have shown that during the pandemic, QoL among youth has been impacted significantly with a decline in average life satisfaction [64]. It has been indicated through evidence that the pandemic led to an increase in paediatric emergency department visits for attempted suicide and suicidal ideation in many countries [65]. Consequently, youth must be considered a priority group for interventions and action in pandemic recovery plans.

Another important finding is that lower-income and ethnocultural minority children reported low-to-moderate QoL. In Canada, many ethnic minority families have been disproportionately affected by the disruption to work compared to non-minorities [66]. Policy actions should focus on extending economic opportunities to disadvantaged populations. Moreover, in addition to addressing other factors such as poor housing and food insecurity, entrenched forms of discrimination should be critically confronted in policy [66].

Children and youth of immigrant parents have also been at a disadvantage during the pandemic. Our findings showed that children aged 8–11 years and 16–18 years whose parents were immigrants were more likely to have low-to-moderate QoL. Migrant parents/caregivers often have fewer resources than Canadian-born caregivers to provide their children with adequate housing or assistance with schoolwork (e.g., language barriers, access to devices/internet at home, and satisfactory study spaces) [67]. It is critical to invest in more educational, financial, and MH programs for children in different school and community settings.

The present study has many strengths. We had a diverse sample of children aged 8–18 years and their parents/caregivers who completed the surveys. Previously validated tools were used to collect data on anxiety, depression, and HRQoL. Our study findings need to be interpreted with a few caveats, however. Due to the cross-sectional design of the study, it limits causal inference; however, this approach was appropriate to achieve desired objectives. We had underrepresentation of lower-income families (<$100,000). When our survey rolled out during the summer, restrictions had eased—empowering children to go outside and bear some semblance of normalcy during the pandemic. This may have skewed responses to questions on feelings and symptoms experienced. The results may also be affected by social desirability bias/non-response bias, e.g., being an online survey, participants who had access to a computer/internet were more likely to take the survey.

Our findings have several significant implications. Our results can provide guidance to policy makers in making decisions, especially when considering the potential link between pandemic effects and mental health in children. These findings also raise awareness among healthcare professionals, parents, and educators about the risk and protective factors of mental health outcomes, allowing for the provision of additional resources and services to support children. Our research has also highlighted the disparities that exist within ethnocultural minority groups. Therefore, studies focusing on understanding the effects of the pandemic on minority populations are warranted. It is imperative for practitioners to exhibit cultural sensitivity and recognize the unique cultural factors that may influence the mental health of children during and after the pandemic.

## 5. Conclusions

Overall, this study provides significant insights into the mental health and health-related quality of life of children and youth during the first year of the pandemic. The results demonstrate the prevalence of children and youth’s MH likely declining during the COVID-19 pandemic compared to before. Sociodemographic, COVID-19-contributing, and behavioural factors determined MH outcomes. However, the cross-sectional nature of the study restricts the ability to establish a causation between the pandemic and mental health outcomes, and self-reported responses on validated tools may yield different results from clinical evaluations. This study calls for coordinated and comprehensive actions to develop targeted interventions and scale up services. Our study also brought to light inequities existing within ethnocultural minority groups. Equity considerations must be a priority at the core of decision making across policy sectors. More longitudinal studies are required to understand the long-term effects of the pandemic on paediatric mental health. Conducting qualitative inquiries could help to comprehend why some children were able to cope better than others and identify factors that can nurture resilience.

## Figures and Tables

**Figure 1 children-10-01009-f001:**
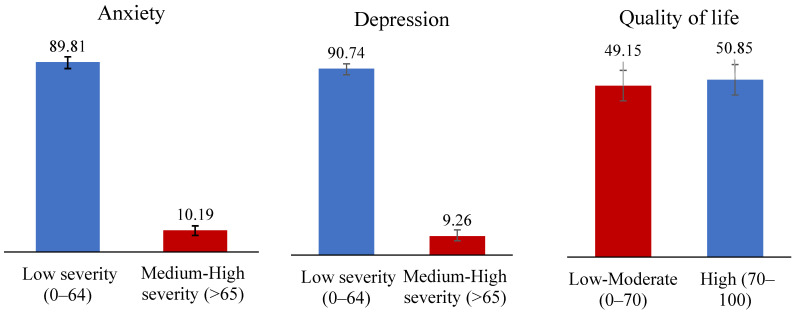
Point Prevalence (Last 2 Weeks) of Anxiety and Depression Symptoms and Health-Related Quality of Life for Children and Youth (8–18 Years).

**Figure 2 children-10-01009-f002:**
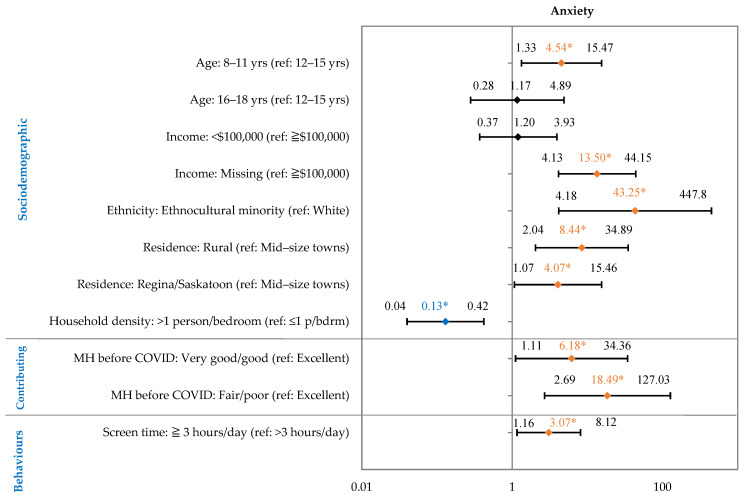
Factors Associated with Self-Reported Medium–High Anxiety Symptoms in Children (8–18 Years) During the First Year of the Pandemic in Saskatchewan, Canada. Forest plot shows main effects: odds ratio shown in diamond, 95% confidence intervals on either side. * Odds ratio in orange indicate a statistically significant risk factor. Odds ratio in blue indicate a statistically protective factor. Full regression model presented in Appendix A.

**Figure 3 children-10-01009-f003:**
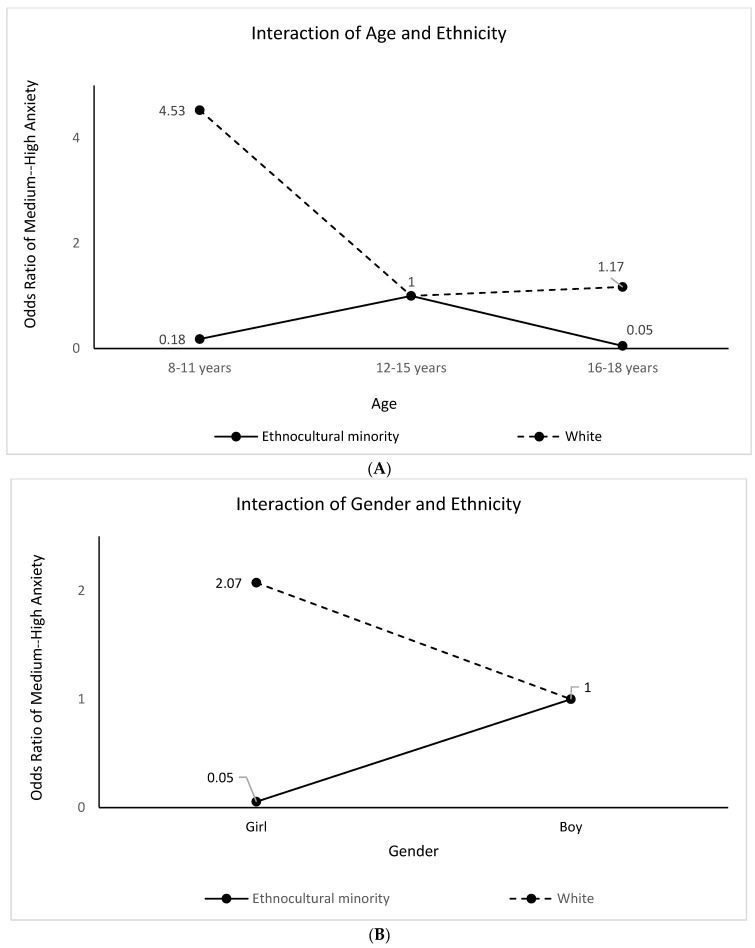
(**A**) Ethnicity Modifies the Effects of Age on Medium–High Anxiety Symptoms. (**B**) Ethnicity Modifies the Effects of Gender on Medium–High Anxiety Symptoms.

**Figure 4 children-10-01009-f004:**
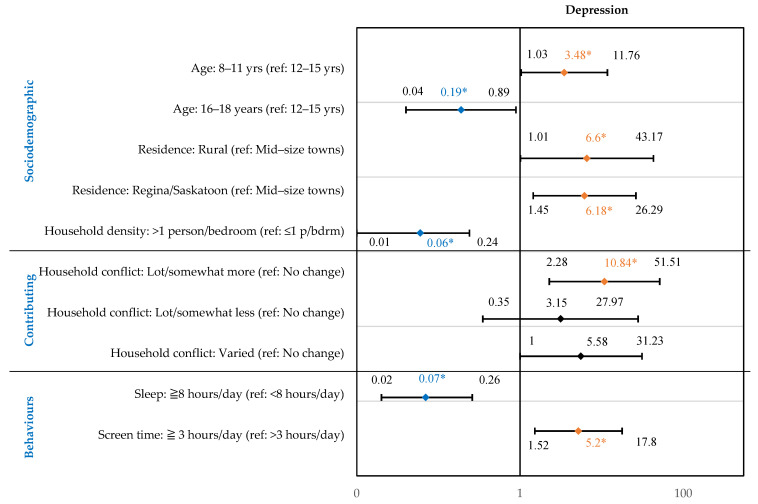
Factors Associated with Self-Reported Medium–High Depression Symptoms in Children (8–18 Years) During the First Year of the Pandemic in Saskatchewan, Canada. Forest plot shows main effects: odds ratio shown in diamond, 95% confidence intervals on either side. * Odds ratio in orange indicate a statistically significant risk factor. Odds ratio in blue indicate a statistically significant protective factor. Full regression model presented in Appendix A.

**Figure 5 children-10-01009-f005:**
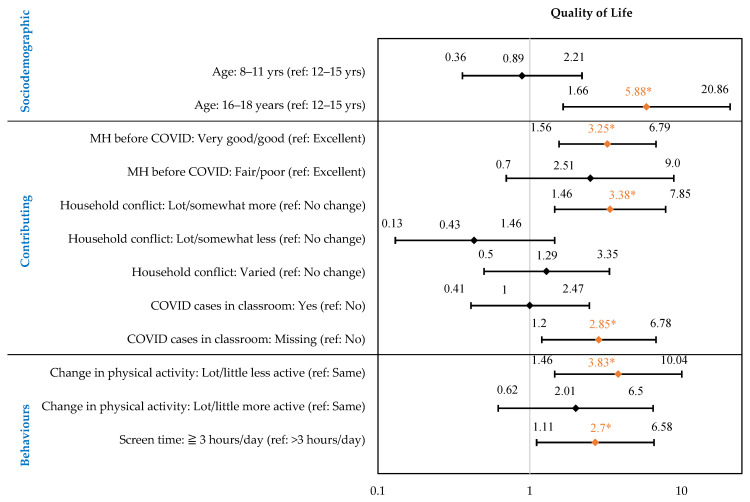
Factors Associated with Self-Reported Low–Moderate Quality of Life in Children (8–18 Years) During the First Year of the Pandemic in Saskatchewan, Canada. Forest plot shows main effects: odds ratio shown in diamond, 95% confidence intervals on either side. * Odds ratio in orange indicate a statistically significant risk factor. Full regression model presented in Appendix A.

**Figure 6 children-10-01009-f006:**
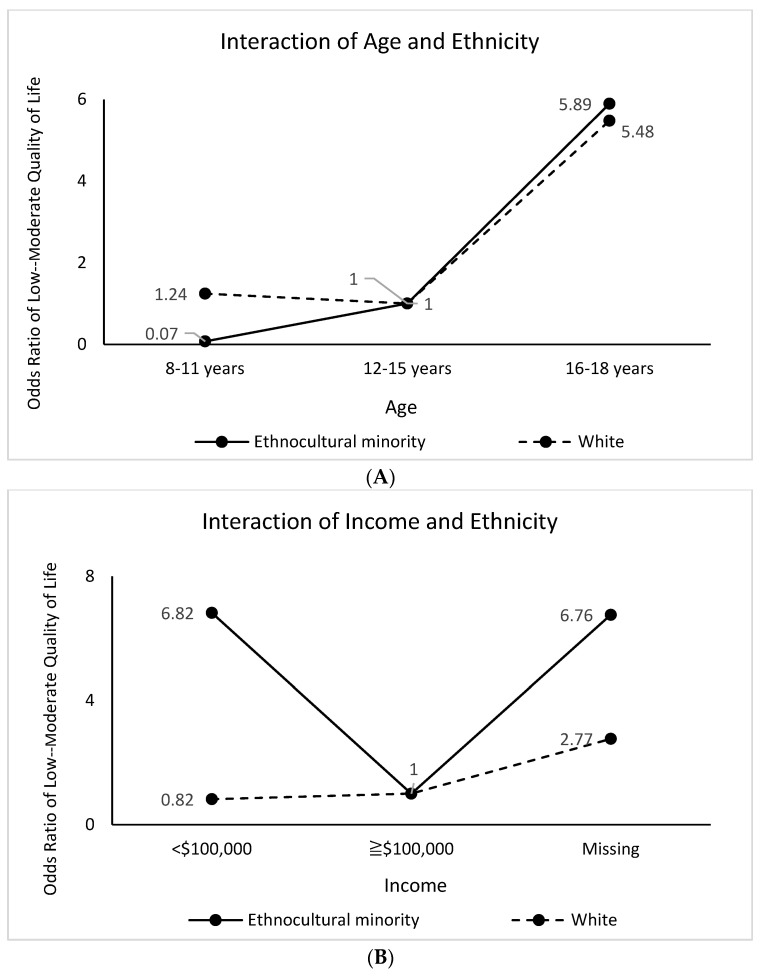
(**A**) Ethnicity Modifies the Effects of Age on Low–Moderate Health-Related Quality of Life. (**B**) Ethnicity Modifies the Effects of Income on Low–Moderate Health-Related Quality of Life. (**C**) Parent Immigration Status Modifies the Effects of Age on Low–Moderate Health-Related Quality of Life.

**Table 1 children-10-01009-t001:** Independent Variables.

Factors	Variables
Sociodemographic	Age: 8–11 yrs, 16–18 yrs, 12–15 yrs
	Grade: Elementary (1–8), High (9–12)
	Gender: girl/woman, boy/man
	Family income: <$100,000, missing, ≥$100,000
	Ethnicity: Ethnocultural minority, white
	Immigration status: Either one parent/none of the parents born in Canada, both parents born in Canada
	Place of residence: Rural, Regina/Saskatoon, Mid-size city/town
	Household density: >1 person/bedroom, ≥1 person/bedroom
Behavioural	Sleep: ≥8 h, <8 h
	Physical activity: <7 days/week MVPA, ≥7 days/week MVPA
	Change in physical activity: Lot/little less active, lot/little more active, no real change
	Recreational screen time: ≥3 h, <3 h
COVID-19-contributing	Mental health status before COVID: Fair/poor, very good/good, excellent
	Household conflicts: Lot/somewhat more, Lot/somewhat less, Varied, No real change
	Schooling situation (mode of learning): Either in-class or online, both in-class and online
	COVID-19 cases in classroom: Yes, missing, no

**Table 2 children-10-01009-t002:** Key Sociodemographic Characteristics (N = 510 Child–Parent Dyads); See Us, Hear Us Study, Saskatchewan, Canada, 2021.

	*n* (%)
Age group (years)	
8 to 11	238 (46.71)
12 to 15	174 (34.12)
16 to 18	98 (19.18)
Grade	
Elementary (1–8)	333 (65.29)
High (8–12)	163 (31.93)
Missing	14 (2.78)
Gender	
Girl/woman	223 (43.72)
Boy/man	260 (50.98)
Missing	27 (5.29)
Income	
<$100,000	184 (36.04)
≥$100,000	203 (39.76)
Missing	123 (24.19)
Ethnicity	
Ethnocultural minority	121 (23.74)
White	335 (65.61)
Missing	54 (10.66)
Place of residence	
Mid-size city/town	165 (32.40)
Rural	85 (16.63)
Regina/Saskatoon	199 (39.01)
Missing	61 (11.96)
Parent immigration status	
Both born in Canada	401 (78.61)
None/either one born in Canada	69 (13.60)
Missing	40 (7.79)

Note: Weighted numbers are presented after rounding.

## Data Availability

The dataset used for this study is available upon request from the first author.

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
