# Peer review of "Mental Health and Health-Related Quality of Life of Children and Youth during the First Year of the COVID-19 Pandemic: Results from a Cross-Sectional Survey in Saskatchewan, Canada"

_children, 2023, doi:10.3390/children10061009_

Round 1
Reviewer 1 Report
The paper reports on the prevalence and determinants of anxiety, depression and experiences of health-related quality-of-life symptoms of children and youth during the first year of the COVID-19 pandemic in Saskatchewan, Canada
The topic is highly relevant for various disciplines, including public health. However, the paper needs improvement on several issues:
1.
The Introduction section is somewhat short considering the number of predictors tested; at least a short literature review on these predictors should be included in the Introduction section.
2.
“The theoretical perspective guiding our study is that relationships between 73 socioeconomic, cultural, behavioural, and social networks influence health outcomes and 74 determine inequities [12,13]”.
The link between these variables can hardly be considered a theoretical perspective. Is there a theoretical perspective employed in the paper?
3.
“Although studies have identified the prevalence of MH difficulties in children 80 during the pandemic, there is still a need for child-reported data and an understanding of 81 the mechanisms and processes that influence MH [16]. Additionally, the impact of 82 COVID-19 and governmental response to it differed across Canadian provinces and 83 territories.”
Does reference 16 and its sentence refer to Canada? Is the last sentence summary of a statement of reference 16, or is it the authors’ view?
4.
Figures 1 and 2 need to have Sources stated below them.
5.
“Our primary objective was to estimate MH effects on children and youth (8-18 86 years) in Saskatchewan across three domains: anxiety, depression, and health-related 87 quality of life to coincide with the end of first fill academic year during COVID-19 and 88 identify factors that increase or decrease risk for these MH outcomes. The second objective 89 was to examine whether these impacts were worsened or buffered among equity- 90 deserving groups (those who experience structural and systemic marginalization, such as 91 minority and racialized communities)”
“..to estimate MH effects on children and..” That does not make much sense. The determinants/factors of MH were examined. Or the effect of the pandemic (although it is difficult to test this) on MH was examined. Please reword.
6.
The term “equity in health” should be defined, as it is often used interchangeably in the literature with the terms health inequalities and health differences.
7.
“The disruption of available 372 MH and counselling supports for children also explains the observed higher MH 373 symptoms in children and youth residing in rural, and urban areas.”
It is unclear how this factor would explain differences in MH in rural/urban areas vs small towns.
8.
“It was not unexpected to find that children and youth reporting increased 377..”
I suggest: It was expected …
9.
The Discussion section should provide more contextualization regarding the study findings and prior literature.
10.
Considering the issues mentioned, I would advise revising and resubmitting.
Please, see above.
Author Response
Reviewer 1
1. The Introduction section is somewhat short considering the number of predictors tested; at least a short literature review on these predictors should be included in the Introduction section.
Response: Literature with studies conducted during the early pandemic have been added to the introduction section including overall prevalence of mental health outcomes seen on a global level and in the Canadian context. Studies on different predictors (sleep, screen time, physical activity, sociodemographic, online schooling) and their association with mental health have been added to provide additional context to the study. Edits are on lines 61-91 (see clean revised document).
2.“The  theoretical  perspective  guiding  our  study  is  that  relationships  between  socioeconomic, cultural, behavioural, and social networks influence health outcomes and determine  inequities  [12,13]”. The link between these variables can hardly be considered a theoretical perspective. Is there a theoretical perspective employed in the paper?
Response: We chose to guide our analysis using the WHO Commission on Social Determinants of Health framework (2004). The CSDH framework explains how the different social determinants of health influence health outcomes and health inequities across populations. This has been added to the introduction section.
3. Response: Ref #16 (now #26) refers to the point stating that there is need to understand specific processes and mechanisms impacting the mental health of children. The sentence “Additionally, the impact of COVID-19 and governmental response to it differed across Canadian provinces and territories. Therefore, a comprehensive understanding of the MH impact on children and youth in each jurisdiction is warranted” refer to the authors’ view.
4. Figures 1 and 2 need to have Sources stated below them.
Response: These are original figures created by the research team.
5. “Our primary objective was to estimate MH effects on children and youth (8-18  years)  in  Saskatchewan  across  three  domains:  anxiety,  depression,  and  health-related quality of life to coincide with the end of first fill academic year during COVID-19 and identify factors that increase or decrease risk for these MH outcomes. The second objective was to examine whether these impacts were worsened or buffered  among  equity- 90 deserving groups (those who experience structural and systemic marginalization, such as minority  and  racialized  communities)”
“..to estimate MH effects on children and..” That does not make much sense. The determinants/factors of MH were examined. Or the effect of the pandemic (although it is difficult to test this) on MH was examined. Please reword.
Response: Edited the objective. “Our primary objective was to identify factors that increase or decrease risk across three MH domains: anxiety, depression, and health-related quality of life in children (8-15 years-old) and youth (16-18 years-old) in Saskatchewan – to coincide with the end of first fill academic year during COVID-19.”
This has been reworded. Edits on lines 98-105.
6. The term “equity in health” should be defined, as it is often used interchangeably in the literature with the terms health inequalities and health differences.
Response: The definition of equity in health has been added to the introduction section - lines 74-76.
7. “The disruption of available MH  and  counselling  supports  for  children  also  explains  the  observed  higher  MH  symptoms in children and youth residing in rural, and urban areas.”
It is unclear how this factor would explain differences in MH in rural/urban areas vs small towns.
Response: Explanation on differences in MH in rural and urban areas as compared to smaller towns is detailed on lines 384-389.
“The lack of available MH and counseling supports for rural citizens in Saskatchewan ​[45]​, and the rising number of COVID-19 cases in cities like Saskatoon and Regina ​[46]​ during that period can help elucidate the observed higher MH symptoms in children and youth residing in both rural and urban areas compared to smaller towns.”
8. “It was not unexpected  to  find  that  children  and  youth  reporting  increased..”
Response: Reworded it to- It was “expected”.
9. The Discussion section should provide more contextualization regarding the study findings and prior literature. 
Response: More literature contextualizing the study findings has been added to the discussion section.
Reviewer 2 Report
1. In the introduction, the authors have established the background to the study. Enough evidence of the research gaps that necessitated the study, as well as the purpose and have been well established. However, before presenting the state of the art in terms of Canadian studies in relation to children and youth during the Covid-19, it would be refreshing to hear summarized findings from other regions to give a global perspective to the background by mentioning the effects of the covid-19 on children and youth in particular countries across the globe and the theoretical projections from such studies.
2. The methods section is scholarly and offers enough comprehensive data for replication of the study.
3. I hope the age categories of youth, children, etc. are substantiated with evidence in the Canadian context. From 8 years as youth?
4. In the concluding section, describe the key limitations of the study that might have affected the results.
5. In the concluding section, suggest further areas of research.
Generally, this is a well-planned, scholarly delivered study with novel findings that have significant implications for policy and practice.
Author Response
Reviewer 2
1. In the introduction, the authors have established the background to the study. Enough evidence of the research gaps that necessitated the study, as well as the purpose and have been well established. However, before presenting the state of the art in terms of Canadian studies in relation to children and youth during the Covid-19, it would be refreshing to hear summarized findings from other regions to give a global perspective to the background by mentioning the effects of the covid-19 on children and youth in particular countries across the globe and the theoretical projections from such studies
Response: This has been addressed in previous reviewer’s comment.
2. The methods section is scholarly and offers enough comprehensive data for replication of the study.
Response: Thank you.
3. I hope the age categories of youth, children, etc. are substantiated with evidence in the Canadian context. From 8 years as youth
Response: We have clarified the use term, ‘children’ (as those 8-15 years-old), and ‘youth’ (16-18 years-old) right within where we state our objectives for the study.
4. In the concluding section, describe the key limitations of the study that might have affected the results
Response: Key limitations to the conclusion section have been added (lines 493-496). This includes 1) the cross-sectional nature of study making it difficult to establish causation and 2) self-reported responses to the tools used making responses different from those clinically evaluated.
5. In the concluding section, suggest further areas of research
Response: Future research directions have been added to the concluding section (lines 499-503).
“More longitudinal studies are required to understand the long-term effects of the pandemic on pediatric mental health. Conducting qualitative inquiries could help to comprehend why some children were able to cope better than others and identify factors that can nurture resilience.”
6. Generally, this is a well-planned, scholarly delivered study with novel findings that have significant implications for policy and practice
Response: Thank you.
Reviewer 3 Report
Dear Authors,
Congratulations on your extensive work, concerning Mental health and health-related quality of life of children and 2 youth during the first year of the COVID-19 pandemic: Results 3 from a cross-sectional survey in Saskatchewan, Canada
I suggest some minor revisions:
M&M
Brief Timeline of COVID-19 Measures, and Schools in Saskatchewan
In my opinion, lines 96-109 sounds more like background, than a material and methods.
Discussion:
The last part of discussion should be dvided into subsections: strenghts and limitations, future research implications, practical clinical implications, and then study conclusions.
In general, a good job!
Author Response
Reviewer 3
M&M
1. Brief Timeline of COVID-19 Measures, and Schools in Saskatchewan
In my opinion, lines 96-109 sounds more like background, than a material and methods.
Response: The reason we wanted to keep this section, Brief Timeline of COVID-19 Measures, and Schools in Saskatchewan, in the Methods section is because it provides specific context to the study, regarding COVID-19 in Saskatchewan and how schools reacted to the pandemic. In that way, this section is like ‘study setting’ that leads directly to a presentation of methods and material used.
2. Discussion:
The last part of discussion should be dvided into subsections: strenghts and limitations, future research implications, practical clinical implications, and then study conclusions.
Response: We have added a paragraph (lines 476-486) describing the significant implications (clinical, policy, research, practical) of our study.
“Our findings have several significant implications. Our results can provide guidance to policy makers in considering the potential link between pandemic effects and mental health in children when making decisions. These findings also raise awareness among healthcare professionals, parents, and educators about the risk and protective factors of mental health outcomes, allowing for the provision of additional resources and services to support children. Our research has also highlighted the disparities that exist within ethnocultural minority groups. Therefore, studies focusing on understanding the effects of the pandemic on minority population is warranted. It is imperative for practitioners to exhibit cultural sensitivity and recognize the unique cultural factors that may influence the mental health of children during and after the pandemic.”
Round 2
Reviewer 1 Report
The paper is now publishable.
The paper is now publishable.